# Mapping Immune Correlates and Surfaceome Genes in *BRAF* Mutated Colorectal Cancers

Esther Cabañas Morafraile [1,2,*], Cristina Saiz-Ladera [2], Cristina Nieto-Jiménez [2], Balázs Győrffy [3,4,5], Adam Nagy [3,4,5], Guillermo Velasco [2,6], Pedro Pérez-Segura [2] and Alberto Ocaña [2,7,*]

1   Center for Biological Research Margarita Salas (CIB-CSIC), Spanish National Research Council, 28040 Madrid, Spain
2   Experimental Therapeutics Unit, Medical Oncology Department, Hospital Clínico Universitario San Carlos (HCSC), Instituto de Investigación Sanitaria San Carlos (IdISSC), 28040 Madrid, Spain
3   Department of Bioinformatics, Semmelweis University, 1094 Budapest, Hungary
4   2nd Department of Pediatrics, Semmelweis University, 1094 Budapest, Hungary
5   TTK Lendület Cancer Biomarker Research Group, Institute of Enzymology, 1117 Budapest, Hungary
6   Department of Biochemistry and Molecular Biology, Complutense University, 28040 Madrid, Spain
7   Centro de Investigación Biomédica en Red en Oncología (CIBERONC), 28029 Madrid, Spain
*   Correspondence: e.c.morafraile@csic.es (E.C.M.); alberto.ocana@salud.madrid.org (A.O.)

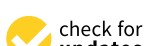



**Simple Summary:** The identification of novel therapeutic strategies for Colorectal Cancer (CRC) patients with *BRAF* mutations is mandatory, since most of the current treatments against this tumor type, including novel compounds, show limited efficacy. In this article, we describe upregulated proteins in the surface of cells within the tumor that can be used as targets to specifically guide novel treatments. In addition, we also observed that the transcriptomic profile matches with antigen presenting cells, such as dendritic cells and macrophages having "antigen processing and presentation of exogenous peptide antigen via MHC class II" as main molecular function, favoring an immunoreactive microenvironment. Therefore, the combination of anti PD(L)1, together with other co-inhibitor receptors from the ones presented in this manuscript, should be explored to treat *BRAF* mutated CRC patients.

**Abstract:** Despite the impressive results obtained with immunotherapy in several cancer types, a significant fraction of patients remains unresponsive to these treatments. In colorectal cancer (CRC), B-RafV600 mutations have been identified in 8–15% of the patients. In this work we interrogated a public dataset to explore the surfaceome of these tumors and found that several genes, such as GP2, CLDN18, AQP5, TM4SF4, NTSR1, VNN1, and CD109, were upregulated. By performing gene set enrichment analysis, we also identified a striking upregulation of genes (CD74, LAG3, HLA-DQB1, HLA-DRB5, HLA-DMA, HLA-DMB, HLA-DPB1, HLA-DRA, HLA-DOA, FCGR2B, HLA-DQA1, HLA-DRB1, and HLA-DPA1) associated with antigen processing and presentation via MHC class II. Likewise, we found a strong correlation between PD1 and PD(L)1 expression and the presence of genes encoding for proteins involved in antigen presentation such as CD74, HLA-DPA1, and LAG3. Furthermore, a similar association was observed for the presence of dendritic cells and macrophages. Finally, a low but positive relationship was observed between tumor mutational burden and neoantigen load. Our findings support the idea that a therapeutic strategy based on the targeting of PD(L)1 together with other receptors also involved in immuno-modulation, such as LAG3, could help to improve current treatments against BRAF-mutated CRC tumors.

**Keywords:** BRAF; colorectal cancer (CRC); immune infiltrates; surface targets; anti-PD(L)1

## 1. Introduction

The identification within many tumors of druggable oncogenic vulnerabilities settled the basis for the development of a new generation of selective therapeutic agents that are

currently in use in the clinical setting [1,2]. One of the strategies that has been used to act on these targets is based on the use of small molecule inhibitors that can permeate cell membranes and inhibit the function of intracellular proteins. Conversely, monoclonal neutralizing antibodies act by blocking soluble factors or plasma membrane proteins containing extracellular domains [3,4]. Examples of successful targeted therapies in cancer include, among many others, the small molecule inhibitors of the BRAF V600 oncogene in melanoma, and monoclonal antibodies against the Epidermal Growth Factor Receptor (EGFR) in non-small cell lung cancer (NSCLC) [1]. Another more recent successful example is the use of inhibitors that can selectively target the K-RAS p.G12C mutation in NSCLC [5,6]. The progressive implementation of these types of selective therapeutic strategies has revolutionized cancer treatment, leading to the development of more individualized therapies that have demonstrated significant improvements in the outcome of patients that harbor specific mutations [4].

Although Colorectal Cancer (CRC) is one of the most prevalent tumors, the identification of druggable molecular alterations lags behind other less frequent solid tumors, such as melanoma or NSCLC, where the use of selective therapies has clearly demonstrated clinical efficacy [7]. Recently, several druggable mutations have been reported in genes frequently mutated in CRC, including those affecting *BRAF* or *K-RAS*, among others [8]. In the case of *K-RAS* mutations, their presence in CRC predicted a lack of response to anti-EGFR antibodies [9].

In CRC, *B-RafV600* mutations have been identified in 8–15% of the patients and were associated with detrimental outcomes and a lack of response to anti-EGFR inhibitors [10]. Treatment with BRAF inhibitors such as Encorafenib has shown activity, particularly if combined with anti-EGFR antibodies [11]. However, although this and other similar combinational therapeutic strategies are promising, only a minor fraction of patients exhibited positive responses to these treatments. Moreover, most of the patients who respond to this therapeutic approach have a limited extension in progression-free survival [10,11]. In addition to this limitation, the toxicity profile of these agents could in some cases impair their administration [12]. In this context, the identification of novel druggable opportunities or therapeutic options to improve activity and reduce toxicity is mandatory.

Lately, immunotherapy has gained momentum by demonstrating impressive clinical responses in various cancer types [13]. Several strategies have been assayed to influence or reactivate the immune system of cancer patients in a manner that can contribute to fight malignant recurrent diseases. One of these strategies is interfering with the inhibitory signal that blocks the host immune response on cancer cells by using anti PD(L)1 antibodies. Another option is the use of antibodies against membrane receptors to boost antibody drug dependent cytotoxicity (ADCC) or complement dependent cytotoxicity (CDC) [14]. An additional approach is the use of antibodies against membrane proteins to vectorize small molecule inhibitors or chemotherapeutic agents as antibody drug conjugates (ADCs) [15,16]. With this approach, off-target non-tumor toxicity will be considerably reduced as the therapy is directly guided to the cancer cell [17].

The surfaceome is considered the set of all plasma membrane proteins that have extracellular domains. In a cancer context, the clarification of the surfaceome of the cell types that form the tumor mass, including cancer and stromal cells, is of paramount interest as it can help to identify specific therapeutic targets and therefore facilitate the design of novel individualized therapies [18].

In this work, we took advantage of transcriptomic data to explore *BRAF*-mutated CRC. Following this strategy, we identified a surfaceome signature that could help to predict the immune status of these tumors and therefore contribute to design more effective therapies against this CRC subtype.

## 2. Materials and Methods

### 2.1. Identification of BRAF Mutations in CRC Patients, Data Collection and Processing

We used data contained at cBioportal (www.cbioportal.org) (accessed on 1 February 2022) [19,20] (TCGA dataset) to explore all mutations in the *BRAF* gene in patients with colorectal cancer. This web resource also provides mutated variants mapped to genomic domains. Protein expression in the cell membrane was identified using the Human Surfaceome Atlas (https://wlab.ethz.ch/surfaceome/) (accessed on 15 March 2022) [21].

### 2.2. Functional Annotation of De-Regulated Genes

We used the publicly available EnrichR online platform (https://maayanlab.cloud/Enrichr/) (accessed on 16 March 2022) [22] to address the gene ontology biological process and molecular function related to each gene set. We represented the most relevant pathways according to their adjusted p-value.

### 2.3. Outcome Analysis

The KM Plotter Online tool [23] (https://kmplot.com/analysis/, last accessed on 20 March 2022), was used to evaluate the relationship between up-regulated genes' expression and clinical outcome in patients with rectum adenocarcinoma. This open access database contains 165 samples and allowed us to investigate Free Progression (FP) and Overall Survival (OS) of up-regulated genes in the rectum adenocarcinoma subtype. False discovery rate (FDR) indicates replicable associations across multiple studies.

### 2.4. Expression Analysis

The analysis comparing the expression level of individual genes in *BRAF* mutated samples compared with wildtype ones was carried out using data from the Cancer Dependency Map (DepMap) portal (https://depmap.org/portal/, accessed on 28 October 2021) for cell lines and the Xena Functional Genomics Explorer web server from the University of California Santa Cruz (https://xenabrowser.net/, last accessed on 25 March 2022) [24] for TCGA colon and rectum adenocarcinoma samples.

### 2.5. Correlation between Gene Expression and Immune Cell Infiltration

To explore the associations between gene expression and immune infiltration cells we used TIMER2.0 (http://timer.cistrome.org/, accessed on 5 April 2022). TIMER provides 4 modules (Gene, Mutation, sCNA and Outcome) to explore the association between immune infiltrates and genomic changes [25]. The gene-correlation module was used to link gene expression with activation of T cell markers.

### 2.6. Correlation with Tumor Mutational Burden and Antigen Load

The total number of Mutect2-identified somatic mutations was used to compute the "mutational burden" in all TCGA colon and rectal cancer patients [26]. Neoantigen load was determined using the NetMHCpan-3.0 [27], as described previously [28]. Finally, Spearman rank correlation coefficients were computed to correlate mutational burden and neoantigen load and gene expression.

### 2.7. Graphical Design

Bars, dot plots, heatmap and volcano plots were represented using GraphPad Prism software (GraphPad Software 9.0.0, San Diego, CA, USA).

## 3. Results

### 3.1. Mapping Transcriptomic Differences in CRC Tumors Based on BRAF Mutated Status

To identify upregulated genes characteristically expressed in CRC *BRAF* mutated tumors, we interrogated open datasets as described in the materials and methods section. Of 396 patients, 62 of them harbored *BRAF* mutations, corresponding to 15.65% of all patients with CRC. Using a threshold fold change (FC) equal or higher than two, we

selected 210 genes that were upregulated and 280 genes that were downregulated when comparing BRAF mutated CRC patients versus *BRAF* wt ones. Among them, 57 genes were upregulated and presented at the surface of the cells according to the Surfaceome Atlas [21].

We displayed in a volcano plot those genes up and downregulated in *BRAF* mutated versus *BRAF* wt CRC tumors (Figure 1A). We observed that downregulated genes were more abundant, being 57.14% of total dysregulated genes, compared to 42.86% of upregulated genes. However, the FC in upregulated genes was higher, having a mean fold change of 3.35 versus 3.16 in the downregulated ones. When focusing only on those highly upregulated, we identified *GP2* (18.35 FC), *CLDN18* (12.68 FC), *AQP5* (9.27 FC) and *TM4SF4* (6.06 FC) as those with the highest overexpression. On the other hand, exceedingly downregulated genes included SLC39A2 (12.50 FC), LY6G6D (10.00 FC), SLC13A2 (6.25 FC), and FOLR1 (5.56 FC) (Figure 1B).

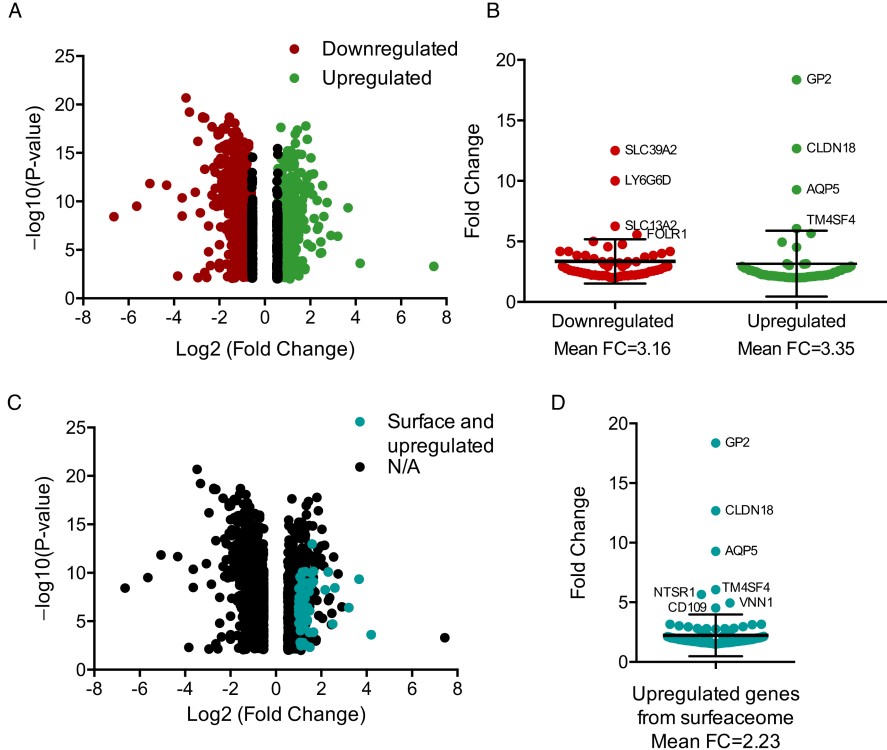

**Figure 1.** Mapping transcriptomic differences in CRC tumors based on *BRAF* mutated status. (**A**). Volcano plot showing statistically significant deregulated genes between BRAF wt and mutant CRC samples, highlighted in color those deregulated genes with an expression decrease (red) or increase (green) equal or higher than 2 folds. (**B**). Dot plot representing the fold change of upregulated and downregulated genes selected in A where error bars are the mean with the standard deviation (SD). (**C**). Same volcano plot as in A, with upregulated and surfaceoma expressed genes in turquoise. (**D**). Dot plot of genes selected in C with error bars representing mean FC and SD.

Upregulated genes that were expressed at the surfaceome are presented in Figure 1C. We noticed that they had a mean FC of 2.23. This group included the previously described genes, in addition to *NTSR1* (5.66 FC), *VNN1* (4.94 FC), and *CD109* (4.52 FC) (Figure 1D). Supplementary Figure S1 displays the expression of the mentioned genes in CRC patients with wt and mutated *BRAF*. In CRC, cell lines AQP5, CLDN18, CD109, and NTSR1 were highly expressed in the *BRAF* mutated group (Supplementary Figure S2).

*3.2. Functional Analysis of Identified Upregulated Genes*

Next, we evaluated the function of the listed upregulated genes. As can be seen in Figure 2A, the main biological functions included the following: "antigen processing and

presentation of exogenous peptide antigen as biological function" and "antigen processing and presentation of exogenous peptide antigen via MHC class II". Genes within these three functions included *CD74, LAG3, HLA-DQB1, HLA-DRB5, HLA-DMA, HLA-DMB, HLA-DPB1, HLA-DRA, HLA-DOA, FCGR2B, HLA-DQA1, HLA-DRB1,* and *HLA-DPA1* (Figure 2A). When evaluating the molecular functions, the most prevalent functions included MHC class II receptor activity and MHC class II protein complex binding, including similar genes (Figure 2B). We noticed that all these genes express proteins that have a strong protein–protein interaction (PPI) (with a PPI enrichment $p$-value $< 1.0 \times 10^{-16}$), forming a cluster of 13 nodes with 64 edges (when the expected number was one), meaning they have an average node degree of 9.85 with an average local clustering coefficient of 0.938 (Figure 2C). This means that these proteins have more interactions among themselves than what would be expected for a random set of proteins. Therefore, such enrichment indicates that the proteins are at least partially biologically connected, as a group.

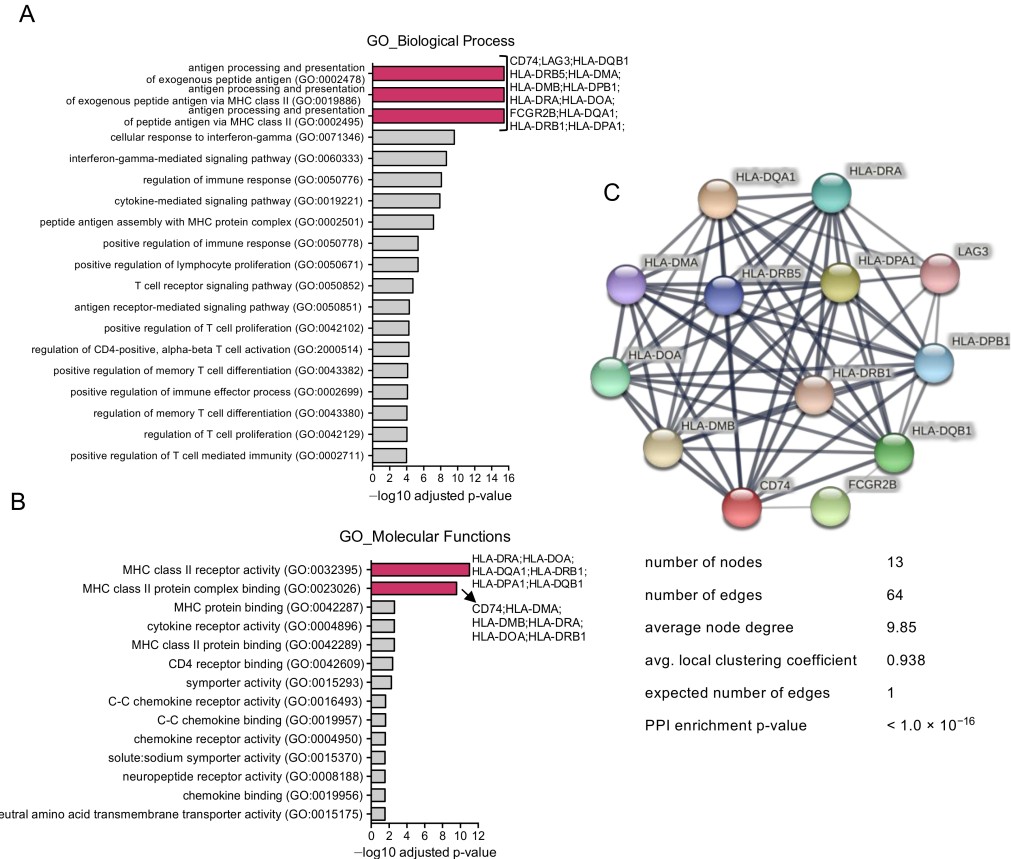

**Figure 2.** Upregulated genes participate in immunological functions. Bar graph showing the top GO biological process (**A**) or molecular functions (**B**) of the selected genes according to their adjusted *p*-value. (**C**). Protein–protein interaction map displaying the significant functional network integrated by the genes involved in the molecular functions and biological process in pink bars in (**A**,**B**).

### 3.3. Surfaceome Immune Related Genes Correlated with Dendritic and T Cell Populations

We next evaluated the association of the genes: *CD74, FCGR2B, HLA-DMA, HLA-DMB, HLA-DOA, HLA-DPA, HLA-DPB1, HLA-DQA1, HLA-DQB1, HLA-DRA, HLA-DRB1, HLA-DRB5,* and *LAG3*, with immune populations including T cell CD8[+], T cell CD4[+] and dendritic cells. We termed this set of genes "CD74 signature". We observed a statistically significant positive correlation between these genes and the mentioned immune populations, especially the antigen presenting cell subgroup represented by the dendritic cells (DC) in colon adenocarcinoma (COAD) (Figure 3 and Supplementary Figure S3). Using another dataset of rectum adenocarcinoma (READ), we observed similar results (Supplementary

Figures S3 and S4). The statistically positive correlation (Rho > 0.5, *p*-value < 0.005) between the selected surfaceome genes and other immune populations is displayed in Supplementary Figure S5, where a positive correlation with macrophages was also observed in the colon adenocarcinoma (COAD) dataset of the rectum. Of note, antigen presentation is also produced by macrophages. At the same time, this set of genes presented a negative correlation with myeloid derived suppressor cells (MSDCs, see Supplementary Figure S5). Finally, we observed that the mentioned CD74 signature correlated with MSI-H tumors and the colon cancer subtype CMS1 that represents 14% of the tumors and is associated with immune strong activation (Supplementary Figures S6 and S7).

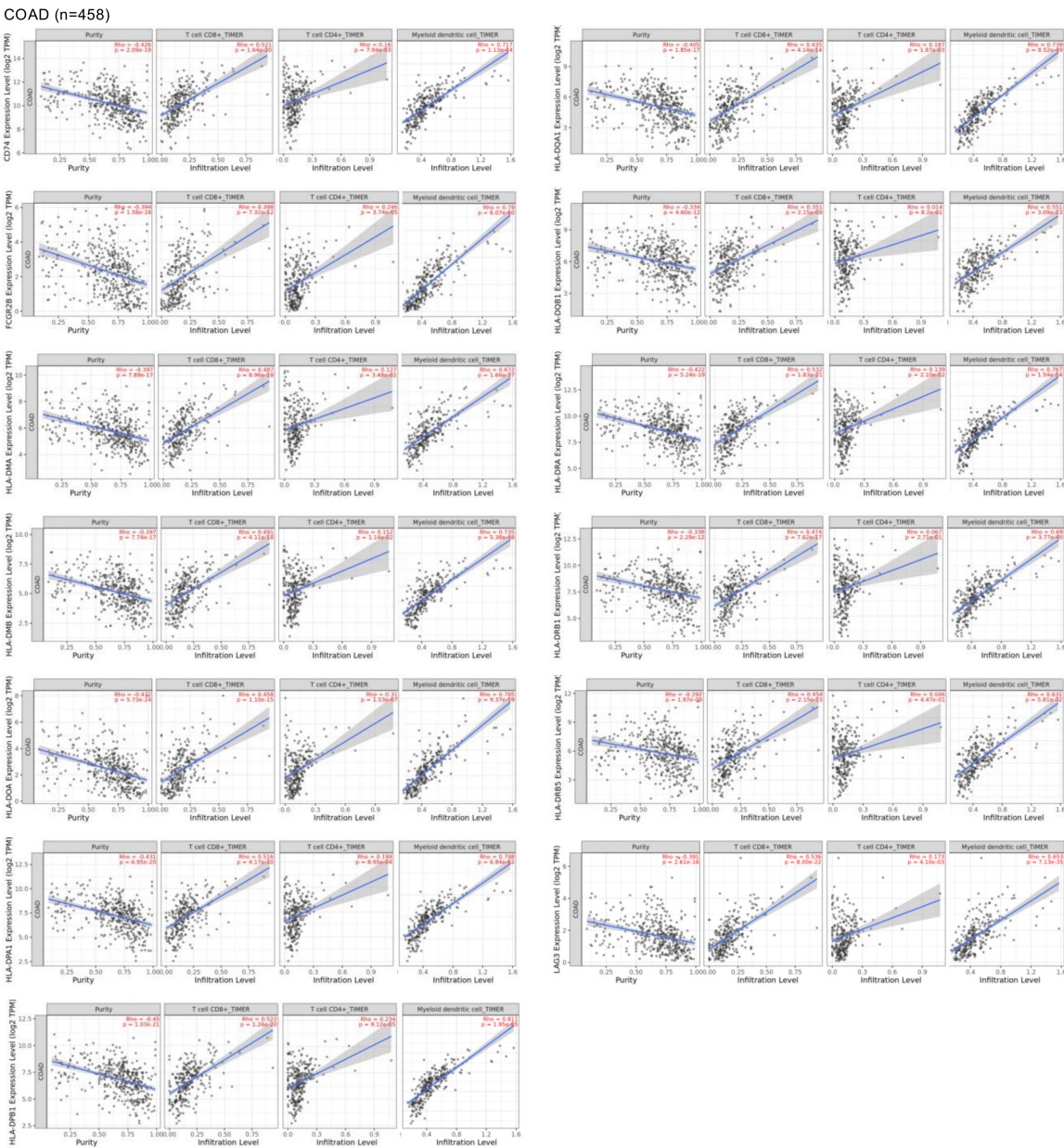

**Figure 3.** Association between the immune surfaceome related genes' expression and the immune cell population tumor infiltrate. Dot plot showing the expression in COAD (n = 458) of upregulated genes (*CD74*, *FCGR2B*, *HLA-DMA*, *HLA-DMB*, *HLA-DOA*, *HLA-DPA1*, *HLA-DPB1*, *HLA-DQA1*, *HLA-DQB1*, *HLA-DRA*, *HLA-DRB1*, *HLA-DRB5*, and *LAG3*) in the Y axes and the infiltration level of some immune cells (CD8+ T, CD4+ T, and myeloids' dendritic cells) in the X axes. The blue regression line shows the positive correlation and Rho and Spearman's *p*-value data are presented in red.

Next, we aimed to correlate the presence of the identified genes with the expression of PD(L)1 or its receptor PD1. As shown in Figure 4, we observed a strong correlation within the presence of the selected genes, particularly for CD74, HLA-DPA1, and LAG3 with PD1 and PD(L)1 in both COAD and READ datasets. These data suggest that an immunosuppressive microenvironment is present with a high expression of exhausted T cells (PD1 positive).

| | COAD (n=458) | | READ (n=166) | |
|---|---|---|---|---|
| | **PD1** | **PD(L)1** | **PD1** | **PD(L)1** |
| *CD74* | Rho=0.747 $p=9.16 \times 10^{-83}$ | Rho=0.712 $p=4.63 \times 10^{-72}$ | Rho=0.74 $p=5.35 \times 10^{-30}$ | Rho=0.569 $p=1.29 \times 10^{-15}$ |
| *FCGR2B* | Rho=0.57 $p=7.69 \times 10^{-41}$ | Rho=0.613 $p=1.55 \times 10^{-48}$ | Rho=0.558 $p=6.04 \times 10^{-15}$ | Rho=0.411 $p=3.75 \times 10^{-08}$ |
| *HLA-DMA* | Rho=0.694 $p=5.28 \times 10^{-67}$ | Rho=0.714 $p=1.66 \times 10^{-37}$ | Rho=0.59 p=6.09 $\times 10^{-17}$ | Rho=0.532 $p=1.54 \times 10^{-13}$ |
| *HLA-DMB* | Rho=0.698 $p=4.89 \times 10^{-68}$ | Rho=0.753 $p=9.18 \times 10^{-85}$ | Rho=0.658 $p=5.58 \times 10^{-22}$ | Rho=0.592 $p=4.27 \times 10^{-17}$ |
| *HLA-DOA* | Rho=0.728 $p=6.51 \times 10^{-77}$ | Rho=0.711 $p=7.7 \times 10^{-72}$ | Rho=0.69 p=8.6 $\times 10^{-25}$ | Rho=0.584 $p=1.49 \times 10^{-16}$ |
| *HLA-DPA1* | Rho=0.726 $p=3.88 \times 10^{-76}$ | Rho=0.761 $p=1.1 \times 10^{-87}$ | Rho=0.669 $p=7.48 \times 10^{-23}$ | Rho=0.608 $p=3.45 \times 10^{-18}$ |
| *HLA-DPB1* | Rho=0.744 $p=4.99 \times 10^{-82}$ | Rho=0.671 $p=3.74 \times 10^{-61}$ | Rho=0.721 $p=7.06 \times 10^{-28}$ | Rho=0.525 $p=3.69 \times 10^{-13}$ |
| *HLA-DQA1* | Rho=0.687 $p=2.97 \times 10^{-65}$ | Rho=0.667 $p=2.82 \times 10^{-60}$ | Rho=0.604 p=7.5 $\times 10^{-18}$ | Rho=0.516 $p=1.11 \times 10^{-12}$ |
| *HLA-DQB1* | Rho=0.5861 $p=1.47 \times 10^{-43}$ | Rho=0.529 $p=2.54 \times 10^{-34}$ | Rho=0.497 $p=9.25 \times 10^{-12}$ | Rho=0.388 $p=2.48 \times 10^{-07}$ |
| *HLA-DRA* | Rho=0.71 $p=2.54 \times 10^{-71}$ | Rho=0.775 $p=6.97 \times 10^{-93}$ | Rho=0.666 $p=1.15 \times 10^{-22}$ | Rho=0.649 $p=3.43 \times 10^{-21}$ |
| *HLA-DRB1* | Rho=0.662 $p=5.79 \times 10^{-59}$ | Rho=0.649 $p=4.72 \times 10^{-56}$ | Rho=0.617 $p=8.71 \times 10^{-19}$ | Rho=0.498 $p=8.29 \times 10^{-12}$ |
| *HLA-DRB5* | Rho=0.598 $p=9.81 \times 10^{-46}$ | Rho=0.577 $p=4.53 \times 10^{-42}$ | Rho=0.574 $p=6.01 \times 10^{-16}$ | Rho=0.483 $p=4.59 \times 10^{-11}$ |
| *LAG3* | Rho=0.831 $p=2.61 \times 10^{-118}$ | Rho=0.738 $p=4.58 \times 10^{-80}$ | Rho=0.787 $p=3.73 \times 10^{-36}$ | Rho=0.635 $p=4.14 \times 10^{-20}$ |

Legend:
- $p<0.005$, Rho >0.5
- $p<0.005$, Rho= 0-0.5
- $p>0.005$
- $p<0.005$, Rho= 0-(−0.5)
- $p<0.005$, Rho <−0.5

**Figure 4.** Positive correlation between identified upregulated genes and PD1/PD(L)1 expression. Heatmap indicating the Rho and Spearman's *p*-value of the correlation between the expression of upregulated genes (*CD74, FCGR2B, HLA-DMA, HLA-DMB, HLA-DOA, HLA-DPA1, HLA-DPB1, HLA-DQA1, HLA-DQB1, HLA-DRA, HLA-DRB1, HLA-DRB5,* and *LAG3)* and PD1 or PD(L)1 in COAD (n = 458) and READ (n = 166).

Of note, the genes that were highly upregulated and described in Figure 1 were not associated with PD (L)1 (Supplementary Figure S8) or any specific immune cell subtype (Supplementary Figure S9). These findings, in addition to the purity score, suggest that these genes correspond to the tumor cells and are not involved in the modulation of the immune system.

*3.4. Immune Related Genes Associate with Clinical Outcome*

The association of the identified genes with clinical outcome was analyzed independently (Figure 5A). As shown in Figure 5B, some genes were associated with detrimental relapse free survival, RFS, including *TNFSF13B, FPR1, HLA-DOA, ITGB7, CD274, HLA-DMB, HLA-DQA1,* and *SLC2A5*, and others with poor overall survival (OS) such as *PTPRU, ABCA3, CX3CL1,* and *VNN1* (Figure 5C). Genes associated with favorable outcomes included *SLC4A4, TM4SF4, GABRP,* and *LYDP5*.

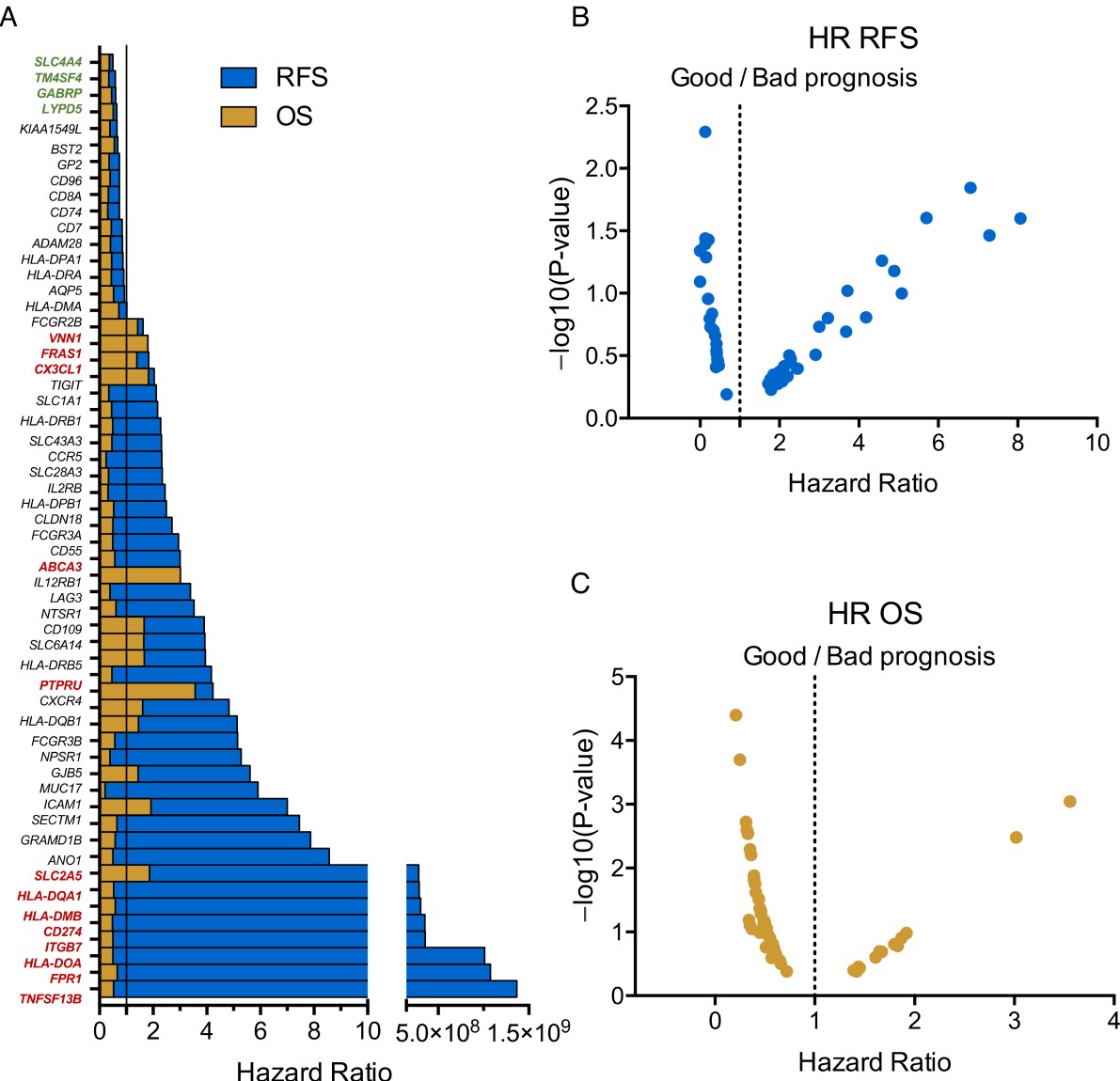

**Figure 5.** Correlation between upregulated genes and prognosis of patients with READ. (**A**) Bar graph showing the Hazard Ratio (HR) in relation to RFS (in blue) and OS (in yellow) of all 57 significantly upregulated genes that belongs to the surfaceome. Line at 1 separates good from bad prognosis. Genes highlighted in green are associated with good prognosis while those in red associate with worst outcome. Vulcano plot representing the *p*-value in Y axes and HR in X ones of RFS (**B**) and OS (**C**) of patients with READ. The dotted line at 1 separates good from bad prognosis.

### 3.5. Correlation with Tumor Mutational Burden and Antigen Load

Then, we integrated the genes by using their mean expression into two signatures termed CD74 and AQP5 signatures and correlated these signatures with the mutation burden (Supplementary Figure S10). In this analysis, both signatures had a low but positive correlation (correlation coefficients (or corr. coeff.) = 0.193, $p = 3.9 \times 10^{-5}$ and corr. coeff. = 0.23, $p = 1.1 \times 10^{-6}$, for the CD74 and AQP5 signatures, respectively). The same analysis was also performed for the neoantigen load by using the number of expressed peptides binding to MHC molecules, and both signatures reached a similar level of significance (corr. coeff. = 0.267, $p = 4.1 \times 10^{-8}$, and corr. coeff. = 0.244, $p = 5.3 \times 10^{-7}$ for the CD74 and AQP5 signatures, respectively).

## 4. Discussion

In the present article, we explored surfaceome genes in CRC patients that harbor mutations in the BRAF gene with the main goal to identify potential immunologic druggable opportunities.

BRAF is an oncogene found to be mutated in a significant proportion of tumors: up to 10–15% of all CRC patients [8]. The BRAF V600 is the most frequent mutation, representing 98% of the mutations in the BRAF gene [8]. This mutation is mainly observed in the right sided CRC, and in poorly differentiated tumors [29]. In addition, tumors with this mutation primarily belong to the consensus molecular subgroup (CMS) CMS1 or microsatellite instability (MSI) immune subtype that integrates microsatellite instability high (MSI H) tumors and associates with an immunologic transcriptomic profile [30]. However, although 42% of BRAF mutated CRC tumors can be included within this group, there is still a high proportion that belong to other subtypes not characterized as immune-related [31].

In our study, we observe that CRC tumors with BRAF mutations express a high number of genes linked with antigen presenting functions. All of them belonged to the major histocompatibility complex (MHC) II molecules that are mainly expressed in professional antigen-presenting cells. Antigens presented in these molecules are extra-cellular proteins that are endocytosed and digested in the lysosomes, and those peptide fragments are exposed by the MHC II molecules [32,33]. Of note, high antigen load correlates with an adaptive immune response and, therefore, with a higher efficacy of immune checkpoint inhibitors to be active [34,35]. Conversely, a high neoantigen load does not guarantee a good response to immune checkpoint inhibitors (ICIs) due to the presence of an immune suppressive microenvironment [35,36].

In line with the previous finding, when matching the immune populations with the transcriptomic profile of BRAF-mutated CRC patients we observed that the best correlation was with antigen presenting cells, such as dendritic cells and macrophages. This finding indicates that BRAF mutated cells favor a microenvironment enriched in antigen presenting cells. In addition, we also observed a moderate correlation with CD8 and a minor correlation with CD4 T cells, and a strong association with PD1 and PD(L)1 expression. The mentioned CD74 signature, that is associated with colon tumors with MSI-H and with the colon cancer subtype termed CMS1 (that represents 14% of all tumors), has been characterized as MSI Immune and it is known to be hypermutated and microsatellite unstable [30]. These findings suggest that, although there is a presence of effector T cells in the tumors, these cells are inhibited or exhausted. Furthermore, some of the genes identified as upregulated in BRAF-mutated CRC patients were associated with detrimental outcomes, including HLA-DOA, CD74, HLA-DMB, and HLA-DQA1, further supporting the idea that the microenvironment of these tumors contributes to the development of an aggressive immunosuppressive phenotype. We therefore postulate the idea that novel strategies aimed at improving the efficacy of ICI in BRAF-mutated CRC patients should combine the use of anti-PD(L)1 antibodies with inhibitors of other receptors involved in the regulation of immune responses that could help to revert this immunosuppressive phenotype. In this context, it is worth underlining the relevance of LAG3 as a negative regulator of effector T cells as well as of antigen presenting cells [37,38]. Interestingly, an anti-LAG3 antibody termed Relatlimab was recently demonstrated to be active in combination with nivolumab as a first line treatment of melanoma [39]. It is therefore tempting to speculate that a similar strategy could be undertaken in BRAF mutated CRC.

In addition to the changes observed in the expression of the immune system-associated genes described above, we also found a strong upregulation of other genes encoding for proteins that contain domains that are exposed at the outer layer of the plasma membrane—including GP2, CLDN18, AQP5, TM4SF4, NTSR1, VNN1, and CD109. The expression of these genes does not correlate with immune populations or PD(L)1 expression, suggesting that their expression is restricted to tumor cells. This finding could also be of great interest since it opens the possibility of using some of these proteins as therapeutic targets against BRAF-mutated CRC. For example, antibodies against CLDN18 (encoding for the tight

junction protein Claudin 18) are currently under development in gastric and pancreatic cancer [40]. Our observations now set the basis for the potential evaluation of these antibodies alone or in combination with anti-PD(L)-1 or anti-LAG-3 antibodies in CRC tumors harboring BRAF mutations. Likewise, VNN1 has been linked with detrimental overall survival [41] and could also be also therapeutically explored.

On the other hand, we acknowledge that our study has limitations. The first one relates to the lack of discrimination between the stromal and tumor compartment. In this context, single cell sequencing could differentiate between the populations where these genes are expressed. Finally, we consider that data presented here should be validated in the laboratory or even in a more accurate manner by using immunohistochemistry (IHC) analysis in human samples.

## 5. Conclusions

Taken together, our findings support the notion that BRAF-mutated colorectal tumors favor a strong immune activated state enriched with antigen-presenting cells and T lymphocytes, and therefore associates with MSI-H and CSM1 CRC tumors. We hypothesize that a therapeutic strategy based on the blockade of PD(L)1, as well as other receptors also involved in immuno-modulation, such as LAG3, together with the pharmacological inhibition of BRAF, could help to improve current treatments against BRAF-mutated CRC tumors.

**Supplementary Materials:** The following supporting information can be downloaded at: https://www.mdpi.com/article/10.3390/curroncol30030196/s1, Figure S1: Bar graph that represents the expression of most upregulated genes (AQP5, CD109, CLDN18, GP2, NTSR1, TMS4F4, and VNN1) comparing samples from BRAF wt and mut COAD patients; Figure S2: Bar graph indicating the expression of most upregulated genes (AQP5, CD109, CLDN18, GP2, NTSR1, TMS4F4, and VNN1) comparing colorrectal BRAF wt and BRAF mut cell lines; Figure S3: Heatmap indicating the Rho and Spearsman's *p*-value of the correlation between the expression of upregulated genes (CD74, FCGR2B, HLA-DMA, HLA-DMB, HLA-DOA, HLA-DPA1, HLA-DPB1, HLA-DQA1, HLA-DQB1, HLA-DRA, HLA-DRB1, HLA-DRB5, and LAG3) and the infiltration level of some immune populations (CD8+ T cells, CD4+ T cells and myeloid dendritic cells (DC) in COAD (*n* = 458) and READ (*n* = 166); Figure S4: Dot plot showing the expression in READ (*n* = 166) of upregulated genes (CD74, FCGR2B, HLA-DMA, HLA-DMB, HLA-DOA, HLA-DPA1, HLA-DPB1, HLA-DQA1, HLA-DQB1, HLA-DRA, HLADRB1, HLA-DRB5, and LAG3) in the Y axes and the infiltration level of some immune cells (CD8+ T, CD4+ T, and myeloids dendritic cells) in the X axes. The blue regression line shows the positive correlation and Rho and Spearsman's *p*-value data are presented in red; Figure S5: Heatmap indicating the Rho and Spearsman's p-value of the correlation between the expression of upregulated genes (CD74, FCGR2B, HLA-DMA, HLA-DMB, HLA-DOA, HLA-DPA1, HLA-DPB1, HLA-DQA1, HLA-DQB1, HLA-DRA, HLA-DRB1, HLA-DRB5, and LAG3) and the infiltration level of some immune population (CD8+ T cells, CD4+ T cells, myeloid dendritic cells, macrophages, and myeloid derived supressor cells (MDSC) in COAD (*n* = 458); Figure S6: Bar graph that represents the expression of CD74 signature in relation to the status of MSI-H, comparing samples from CRC MSI low (0) and CRCMSI high (1). Mann-Whitney statistical p-value ($p = 3.6 \times 10^{-6}$) was performed with a ±95% confidence interval; Figure S7: Bar graph that represents the expression of CD74 signature in each CRC subtypes (CMS1, CMS2, CMS3 and CMS4). Kruskal-Wallis statistical p-value was calculated ($p = 9.8 \times 10^{-28}$) with a ± 95% confidence interval; Figure S8: Dot plot showing the expression in COAD (*n* = 458) (A) and in READ (*n* = 166) (B) of most upregulated genes (AQP5, CD109, CLDN18, GP2, NTSR1, TMS4F4, and VNN1) in the X axes and PD1 or PD(L)1 in the Y axes. The blue regression line shows the positive correlation and Rho and Spearsman's *p*-value data are presented in red; Figure S9: Heatmap representing the Rho and Spearsman's *p*-value of the correlation between the expression of most upregulated genes (AQP5, CD109, CLDN18, GP2, NTSR1, TMS4F4, and VNN1) and the infiltration level of some immune populations (CD8+ T cells, CD4+ T cells, myeloids dendritic cells, macrophages, and myeloid derived supresor cells) in COAD (*n* = 458) and READ (*n* = 166); Figure S10: Heatmap representing the Rho and Spearsman's p-value of the correlation between the expression of most upregulated genes (AQP5, CD109, CLDN18, GP2, NTSR1, TMS4F4, and VNN1) and the infiltration level of some immune populations (CD8+ T cells, CD4+ T cells, myeloids dendritic cells, macrophages, and myeloid derived supresor cells) in COAD (*n* = 458) and READ (*n* = 166).

**Author Contributions:** A.O. conceived the study and E.C.M. undertook the original design of analysis. E.C.M., C.S.-L. and C.N.-J. searched the data and performed the analyses. A.N. and B.G. provided bioinformatic assistance in data search and analysis. G.V., P.P.-S. provided support in data interpretation. A.O. and E.C.M. wrote the manuscript. All authors reviewed, included modification and approved the final version of the manuscript. All authors have read and agreed to the published version of the manuscript.

**Funding:** Alberto Ocaña's lab is supported by the Instituto de Salud Carlos III (ISCIII, PI19/00808), CRIS Cancer Foundation, ACEPAIN, and Diputación de Albacete. Balázs Győrffy's lab is supported by the 2020-1.1.6-JÖVŐ-2021-00013 and 2020-4.1.1.-TKP2020 grants of the NKFIH, Hungary. Work in G. Velasco's group is supported by the Instituto de Salud Carlos III (ISCIII) and co-funded by the European Regional Development Fund (ERDF), "A way to make Europe", grant number grant PI21/00343 integrated into the State Plan for R&D + I 2021-2024. Atanasio Pandiella's lab is funded by the Ministry of Economy and Competitiveness of Spain (BFU2015-71371-R and FEDER). The CRIS Cancer Foundation supports both research groups. Esther C. Morafraile is supported by a "Juan de la Cierva Incorporación" contract of the Spanish Ministry of Science and Innovation with Ref. IJC2019-041728-I.

**Institutional Review Board Statement:** Given the fact that our study did not involve the use of human samples, as it was a bioinformatic study, no ethical approval was required.

**Informed Consent Statement:** Patient consent was waived due to the fact that no human samples were used in the study: Public available bioinformatic data was used.

**Data Availability Statement:** All the data used in this manuscript is open and would be shared upon request to the corresponding authors.

**Acknowledgments:** We would like to thank our funding institutions, the Instituto de Salud Carlos III, the Diputación de Albacete, the AECC Foundation, the Ministry of Science and Innovation of Spain, the Spanish Cancer Centers Network Program, and the European Commission as the FEDER funding program responsible institution. We would like to especially thank our supporting foundations, such as the "Asociación Costuras en la Piel en Apoyo a la investigación de cáncer en Albacete" (ACEPAIN) (http://www.acepainalbacete.es/, accessed on 15 February 2022) and the Cancer Research Innovation Spain (CRIS) (https://criscancer.org, accessed on 13 February 2022), for their continuous efforts to support our work. The authors wish to acknowledge the support of ELIXIR Hungary (www.elixir-hungary.org, accessed on 7 February 2022).

**Conflicts of Interest:** Alberto Ocaña was a full employee of Symphogen, Denmark, from June 2021 to June 2022. Alberto Ocaña is a consultant for Servier. No conflict of interest to declare in relation to this manuscript.

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
