# Peer review of "Mapping Immune Correlates and Surfaceome Genes in BRAF Mutated Colorectal Cancers"

_curroncol, doi:10.3390/curroncol30030196_

Round 1

Reviewer 1 Report

This manuscript reports a strategy based on the mining of publicly available databases to identify significantly upregulated or downregulated surfaceome genes in BRAF mutated colorectal cancer samples. Interestingly, among the most upregulated ones, genes associated with antigen processing and presentation via the MHC-II complex have been identified, many of which associated with poor prognosis in corresponding CRC patients.

There are no particular criticisms of the proposed strategy and, although the conclusions reached by the authors clearly need further studies to constitute a solid rationale for proposing new therapeutic approaches in BRAF mutated colorectal cancer (i.e., anti-LAG3 or anti-CLDN18 antibodies), the results presented may constitute a valid starting point.

Figure 4 is difficult to read: it is advisable to enlarge it or to increase its resolution.

Author Response

Response to Reviewer 1 Comments

Point 1: Figure 4 is difficult to read: it is advisable to enlarge it or to increase its resolution.

Response 1: 

We agree with the reviewer and we have enlarged and increased Figure 4 resolution.

Reviewer 2 Report

Esther C. M., and colleagues present an array of gene analysis across multiple publicly available colorectal cancer databases. Their focus on surfaceome and immune correlation is interesting. However, in best practice, the scRNA-seq, or mass cytometry database should be used instead of bulk RNA-seq, especially for the immune cell correlation. And the paper is not so well written and organized, which is hard to understand completely. Here are some of my raising concerns and questions:

Majors:

·      Overall, authors do not put efforts on BRAF mutation burden with immune or surfaceome, which according to the title they should. In brief, why we need study the BRAF mutated but not other mutation in colorectal cancer.

·      The section 3.1 should belongs to the method part or combine with 3.2. The necessity for introduce the workflow is redundant.

·      Could we apply the conclusion from this paper to colorectal cancer in general or BRAF mutated colorectal cancer specifically? If the answer is no, what is the difference between BRAF mutated vs Non-mutated.

Minors:

·      The title of figure legend should be conclusion, instead of” Gene selection”

·      The BRAF mutation type should be well described, V600E or other mutations?

·      The PD1/L1 expressed on tumor cells or stromal component?

Author Response

Response to Reviewer 2 Comments

Point 1: Overall, authors do not put efforts on BRAF mutation burden with immune or surfaceome, which according to the title they should. In brief, why we need study the BRAF mutated but not other mutation in colorectal cancer.

Response 1: 

Response 1: We appreciate this very interesting suggestion. We focused on the study on BRAF mutations because they are the only ones for which a therapeutic approach exists, Encorafenib, that has received FDA approval and is prescribed by physicians. Other mutations exist and they are used in the clinic as biomarkers: K-RAS mutation, for no administration of anti EGFR therapies, but it has not a therapy approved.

Point 2: The section 3.1 should belongs to the method part or combine with 3.2. The necessity for introduce the workflow is redundant.

Response 2: 

We appreciate the comment of the reviewer. We have removed the Flow Chart in Figure 1 and combine sections 3.1 and 3.2 to facilitate its lecture.

Point 3: Could we apply the conclusion from this paper to colorectal cancer in general or BRAF mutated colorectal cancer specifically? If the answer is no, what is the difference between BRAF mutated vs Non-mutated.

Response 3: 

The conclusion from this paper apply indeed to BRAF mutated CRC specifically, we are sorry for the misunderstanding. We have written in the first line of section 5 “Taken together our findings support the notion that BRAF-mutated colorectal tumours…” (paragraph line 732) and we have added “when comparing BRAF mutated CRC patients versus BRAF wt ones” (paragraph line 231-232) in section 3, to clarify that our analysis finds differences specifically in BRAF mutated CRC tumours.

Point 4: The title of figure legend should be conclusion, instead of  ”Gene selection”

Response 4: 

We agree with the reviewer and we have changed the titles of figure legends to be more conclusive:

- Figure 1. Mapping transcriptomic differences in CRC tumors based on BRAF mutated status

- Figure 2. Upregulated genes participate in immunological functions

- Figure 3. Association between the immune surfeome related genes expression and the immune cell population tumour infiltrate

- Figure 4. Positive correlation between identified upregulated genes and PD1/PD(L)1 ex-pression.

 Point 5: The BRAF mutation type should be well described, V600E or other mutations?

Response 5: 

We appreciate the comment of the reviewer. We refer to any mutation in BRAF gene (including, but not exclusively, V600E). In order to explain it better in the text we have added in section 2: “…to explore all mutations in BRAF gene in patients with Colorectal Cancer.” (paragraph line 168-169).

Point 6: The PD1/L1 expressed on tumor cells or stromal component?

Response 6: 

We agree with the reviewer that it would be an interesting point to address. The data we have used are not single cell sequencing so we cannot make this distinction. However, we have compared data from bulk tumours (that have stromal component and tumour cells) in BRAF mutated and wildtype Colorectal cancer.
